# Constructing artificial life and materials scientists with accelerated AI using Deep AndersoNN

**Saleem Abdul Fattah Ahmed Al Dajani** [1]    **David E. Keyes** [1]

## Abstract

Deep AndersoNN is a framework for accelerating AI by taking the continuum limit as the number of explicit layers in a neural network approaches infinity, and can be taken as a single implicit layer, known as a deep equilibrium model. Solving for parameters of a deep equilibrium model reduces to a nonlinear fixed point iteration problem, enabling use of vector-to-vector iterative solvers and windowing techniques, such as Anderson extrapolation, for accelerating convergence to the fixed point deep equilibrium. Here we show that Deep AndersoNN achieves up to an order of magnitude of speed-up in training and inference. The method is demonstrated on density functional theory results for industrial applications by constructing artificial life and materials 'scientists' capable of classifying biomolecules, drugs, and compounds as strongly or weakly polar, sorting metal-organic frameworks by pore size, and classifying crystalline materials as metals, semiconductors, and insulators, using graph images of node-neighbor representations transformed from atom-bond networks. Results exhibit accuracy up to 98% and showcase synergy between Deep AndersoNN and machine learning capabilities of modern computing architectures, e.g. GPUs, for accelerated computational life and materials science by quickly identifying structure-property relationships. This paves the way for saving up to 90% of compute required for AI, reducing its carbon footprint by up to 60 gigatons per year by 2030, and scaling above memory limitations of explicit neural networks in life and materials science, and beyond.

[1]Applied Physics (AP) Program and Extreme Computing Research Center (ECRC), Physical Science and Engineering (PSE) and Computer, Electrical, Mathematical Sciences and Engineering (CEMSE) Divisions, King Abdullah University of Science and Technology (KAUST), Thuwal, Makkah Province, Kingdom of Saudi Arabia (KSA) 23955-6900. Correspondence to: Saleem A. Al Dajani <saleem.abdulfattah.aldajani@gmail.com>.

*Accepted at the 1st Machine Learning for Life and Material Sciences Workshop at ICML 2024.* Copyright 2024 by the author(s).

## 1. Introduction & Background

High-performance computing (HPC) is becoming essential to artificial intelligence (AI) in the modern paradigm of machine learning (Schwarz, Nicholas et al, 2020). Foundation models, large language models (LLMs), and multi-agent natural language societies of mind (NLSOMs) (Zhuge, Mingchen et al., 2023) require significant computing resources and large amounts of data to achieve practical accuracies with up to trillions of parameters using explicit neural networks (Andrae, Anders S.G. and Edler, Tomas, 2015; de Vries, Alex, 2023; Patterson, David et al., 2021; Jones, Nicola et al., 2018). As the number of layers in a neural network approaches infinity, these models can be approximated with single-layer implicit models, known as deep equilibrium (DEQ) models (Bai, 2022; Bai, Shaojie and Kolter, J Zico and Koltun, Vladlen, 2019; Bai, Shaojie and Koltun, Vladlen and Kolter, J Zico; 2021; Huang et al., 2021; Geng, Zhengyang and Zhang, Xin-Yu and Bai, Shaojie and Wang, Yisen and Lin, Zhouchen, 2021). Solving for the parameters of a single implicit layer that takes both the input, $x$, and the output, $y$, as inputs are reduced to a fixed point iteration problem that is proven to converge to a deep equilibrium state with stable behavior under optimal hyperparameters where the fixed point converges. Vector-to-vector iterative solvers such as Anderson extrapolation (Anderson, 1965; 2019; Ouyang, Wenqing et al., 2020; Fung, Samy Wu et al., 2022) can be employed to accelerate convergence. By accelerating convergence with a window of iterates, performance similar to explicit networks can be achieved while scaling neural networks and reducing the necessary compute resources. Using accelerated DEQ models enables previously computationally prohibitive applications. Here we demonstrate constructing artificial life and materials scientists using Deep AndersoNN, a system to develop scalable machine learning models with Anderson-accelerated DEQ models.

Drug discovery is at the intersection of life and materials science. Density functional theory (DFT) is currently at the forefront of materials modeling, yet suffers from computational limitations with high-atom biological systems needed in the life sciences. Responding to the COVID-19 pandemic required high-throughput, rapid methods to screen

and discover drugs that could be pipelined into laboratory and, ultimately, clinical trials to treat different and evolving variants of the virus (Clyde et al., 2021b;a; Bhati et al., 2021; Clyde et al., 2023; Saadi et al., 2021; Babuji et al., 2020; Lee et al., 2021). Machine learning and scaling DFT enables high-throughput classification of candidate drugs based on their material properties, such as pore size for DNA/RNA capture and dipole moment for polarity.

---

**Algorithm 1** Extrapolation for Fixed Point Iteration (Zico Kolter, David Duvenaud, and Matt Johnson)

---

**Input:** Function $f$, initial guess $x_0$, window size $m = 5$, regularization $\lambda = 1e - 5$, maximum iterations $max\_iter = 1000$, tolerance $tol = 1e - 2$, mixing parameter $\beta = 1.0$

Initialize number of data points $n$, batch size $b$, number of input and output channels $d$, frame height $H$, frame width $W$ from $x_0.shape$

$X, F \leftarrow$ Initialize iterate and function tensors based on $b, m,$ and $d \times H \times W$

$H, y \leftarrow$ Initialize $H$ and $y$ for the least squares solver where $H = G^T G + \lambda I$ in Eqn. 4

$times, res \leftarrow$ Initialize lists for timing and residuals

**for** $k = 2$ **to** $max\_iter$ **do**

    Start timing this iteration

    $n \leftarrow \min(k, m)$

    $G \leftarrow F[:, : n] - X[:, : n]$

    Update $H$ matrix using $G$ in Eqn. 2

    Solve linear system to find $\alpha$ in Eqn. 4

    Update $X$ and $F$ using $\alpha, m,$ & $\beta$ according to Eqn. 5

    Compute residual, $\frac{\|f(z^k, x) - z^k\|_2}{\|f(z^k, x)\|_2 + \lambda}$

    Store time and residual

    Check for convergence

    **if** residual $< tol$ **then**

        **break**

    **end if**

**end for**

**return** $X[:, k\%m](x_0)$, residuals, times

---

## 2. Methods & Datasets

This study showcases Deep AndersoNN with three high-throughput DFT datasets, integrating life and materials science: QMugs, OQMD, and QMOF. QMugs is a dataset of hundreds of COVID-related drug structures (Isert, Clemens et al., 2022) along with their material and biological properties that are a subset of a larger ChEMBL database (Gaulton, Anna et al., 2012). OQMD is a growing database of >1 million compounds from Chris Wolverton's group at Northwestern computed with DFT (Saal, James E et al., 2013; Kirklin, Scott et al., 2015; Shen, Jiahong et al., 2022). QMOF is a dataset of 20,000 metal-organic frameworks (MOFs) from Rosen et al. (Rosen, Andrew S et al., 2022). Each dataset

provides crystal structures along with material and functional properties, where dipole moment, pore size, and band gaps are taken as examples.

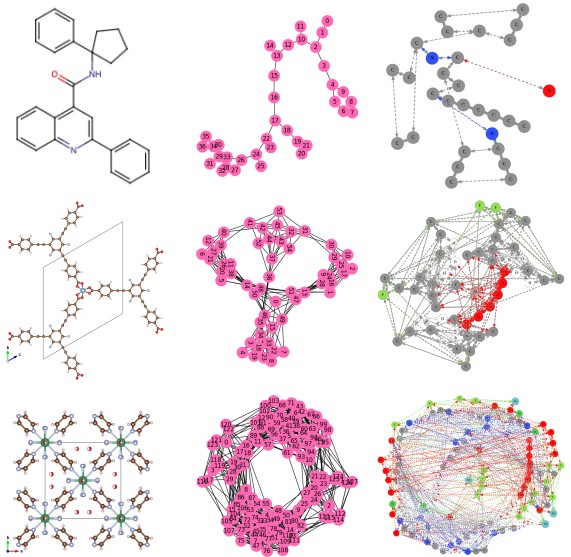

*Figure 1.* Representative SARS-CoV drug compound (top left), representative hypothetical MOF (middle left), and representative experimentally validated MOF (bottom left) molecular structure. Graphical representations of the compounds (center). Node-neighbor representations (right). The node-neighbor representations are inputs for machine learning density functional theory physical properties.

Across each dataset, properties are read with standard libraries for reading comma-separated text files. Labels are deduced by imposing physical limits for these properties, i.e., size of molecule capture for pore size fit, size of band gap for classifying metallic, semiconducting, and insulating behavior, and dipole moment magnitude relative to a one proton-electron dipole moment for polarity. Structures are read as graph images, where structure files in CIF, PDF, SDF, etc. Formats are taken as atom-bond graphs which are then converted to atom node-neighbor representations with a 3.4Å cutoff for MOFs and 2Å cutoff for compounds, then saved as images with each atom as a unique color. An example is shown in Fig. 1.

Fixed-point acceleration begins with the traditional fixed point iteration formula $z^\star = f(z^\star, x)$, which seeks a point $z^\star$ that remains unchanged when a function $f$ is applied to it. Forward iteration, an essential technique for navigating towards this fixed point, is denoted by $z^{k+1} = f(z^k, x)$. This represents the step-wise movement from an initial guess $z^k$ towards the fixed point.

Anderson acceleration refines this process by incorporating a linear combination of prior iterates. This is defined mathematically as $z^{k+1} = \sum_{i=1}^{m} \alpha_i f(z^{k-i+1}, x)$. The weights or

*Table 1.* Comparison of accuracy across four use cases by algorithm, standard versus accelerated, training and testing.

|  | Algorithm | CIFAR10 benchmark | Drug dipole moment | MOF pore sizes | Material band gaps |
|---|---|---|---|---|---|
| Training | Standard | 65% | 77% | 96% | 98% |
|  | Accelerated | 96% | 81% | 97% | 98% |
| Testing | Standard | 64% | 78% | 96% | 98% |
|  | Accelerated | 79% | 80% | 96% | 98% |

coefficients $\alpha_i$ are optimized to minimize the residual vector norm, $\frac{\|f(z^k,x)-z^k\|_2}{\|f(z^k,x)\|_2+\lambda}$, leading to a more rapid convergence than simple standard forward iteration. This optimization is subject to a constraint that ensures the coefficients sum to unity:

$$\text{minimize}_\alpha \quad \|G\alpha\|_2^2, \quad \text{subject to} \quad 1^T\alpha = 1 \quad (1)$$

Given the optimization problem, the matrix $G$ in the minimization problem is composed as follows, where $G$ is defined as:

$$G = \begin{bmatrix} f(z^k,x) - z^k, \cdots, (z^{k-m+1},x) - z^{k-m+1} \end{bmatrix} \quad (2)$$

and $z^k$ represents the solution estimate at iteration $k$, $f$ represents the fixed-point function, $x$ is an input parameter, and $m$ is the memory of past iterates considered in the Anderson extrapolation.

The Lagrangian $L(\alpha,\nu)$ that incorporates the equality constraint into the optimization problem is given by:

$$L(\alpha,\nu) = \|G\alpha\|_2^2 - \nu(1^T\alpha - 1) \quad (3)$$

where $\alpha$ is a vector of coefficients that we are optimizing over, and $\nu$ is the Lagrange multiplier associated with the constraint $1^T\alpha = 1$.

To solve for the coefficients $\alpha_i$, we set up and solve a linear system, where $H = G^TG + \lambda I$:

$$\begin{bmatrix} 0 & 1^T \\ 1 & H \end{bmatrix} \vec{y} = \begin{bmatrix} 0 & 1^T \\ 1 & G^TG + \lambda I \end{bmatrix} \begin{bmatrix} \nu \\ \alpha \end{bmatrix} = \begin{bmatrix} 1 \\ 0 \end{bmatrix} \quad (4)$$

The Anderson acceleration can also incorporate a mixing parameter $\beta$, which allows for a balance between the contributions of the original and extrapolated iterates:

$$z^{k+1} = (1-\beta)\sum_{i=1}^{m}\alpha_i z^{k-1+1} + \beta\sum_{i=1}^{m}\alpha_i f(z^{k-i+1},x) \quad (5)$$

The function $f$, which represents the forward pass through the implicit DEQ layer, ensures that despite the intermediate expansion of the channel depth to $k_1$ inner channels, the output tensor $Z$ retains the same dimensions as the input tensor $X \in R^{n \times d \times H \times W}$ by setting the number of outer channels $k_2$ to number of input and output channels, $d$. This design maintains spatial dimensions across the network layer while enhancing feature representations through depth adjustments and enabling residual learning.

Our pipeline implements Anderson acceleration as the solver for the forward pass of a DEQ model with gradients only calculated for the backward pass. The problem is posed as supervised learning image classification where density functional theory-calculated properties are learned within the implicit layer parameters. We use PyTorch on GPUs.

## 3. Results

The fundamental tradeoff between accuracy, represented by relative residual, $\frac{\|f(z^k,x)-z^k\|_2}{\|f(z^k,x)\|_2+\lambda}$, and time is shown in Fig. 2. This shows the acceleration obtainable for a single inference by a forward pass of a random input $x$ through the DEQ model.

Tuning accuracy is shown in the upper right panel of Fig. 2 by optimizing the window size, $m$, and the iterate-extrapolate mixing ratio, $\beta$, to achieve an order of magnitude higher accuracy. The usage of these parameters is demonstrated in Alg. 1. A representative loss function of Anderson acceleration compared to standard forward iteration is shown in the lower panel of Fig. 2, demonstrating significant speedup to lower loss, with accuracy and speedup results from all runs shown in Tab.1 and Tab. 2.

The behavior of Anderson acceleration in its convergence to the fixed-point, deep equilibrium state are shown in Fig. 3. The upper left panel shows an order of magnitude speedup with Anderson acceleration in comparison to the unstable behavior of standard forward iteration until slow convergence, with fixed learning rate for an accurate comparison. The upper right panel shows highest accuracy, lowest speedup behavior, achieving up to 98% accuracy using both Anderson acceleration and forward iteration, and 4x speedup with Anderson. The lower left panel shows trapping of forward

*Table 2.* Summary of algorithmic speedup of training across four use cases, accelerated over standard.

|  | Algorithm | CIFAR10 benchmark | Drug dipole moment | MOF pore sizes | Material band gaps |
|---|---|---|---|---|---|
| Training time [seconds] | Standard | $1.2 \times 10^4$ | $8.4 \times 10^2$ | $5.3 \times 10^3$ | $2.3 \times 10^3$ |
|  | Accelerated | $1.4 \times 10^3$ | $5.1 \times 10^1$ | $3.0 \times 10^2$ | $5.3 \times 10^2$ |
| Speed-up relative to standard | Ratio | 8.6 | 16.5 | 17.6 | 4.4 |
|  | Compute saved | 88% | 94% | 94% | 77% |

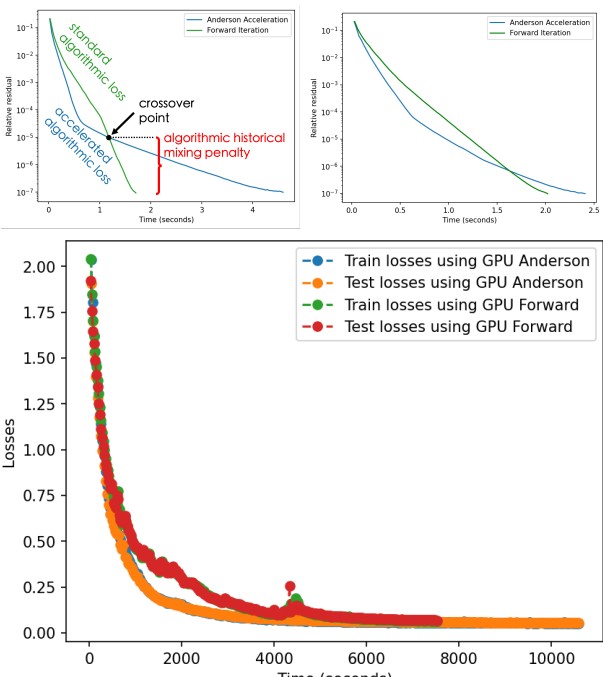

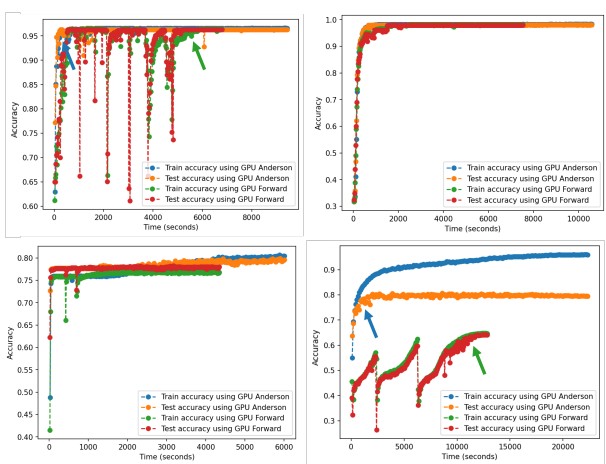

*Figure 2.* Accelerating convergence to fixed point deep equilibrium with (top right) inferences (single forward pass), (top left) representative behavior of speedup tuning for higher accuracy with window size, $m$, and iterate-extrapolate ratio, $\beta$, and (bottom) training until residual tolerance achieved.

*Figure 3.* Achieving accuracy near unity with data augmentation (top left, QMOF pore size classification, and right, OQMD QMOF band gap classification) versus trapping in local minimum by standard forward iteration (lower left, QMugs COVID drug polarity classification), compared to CIFAR10 benchmark (lower right).

iteration in a local minimum with increasing accuracy for Anderson acceleration. The lower right panel shows higher generalization due to lower initialization error for Anderson compared to forward iteration.

Representative confusion matrices are shown in Fig. 4, constructing artificial scientists capable of classifying compounds, drugs, and MOFs based on example structure-property relationships. The summary of these results for all cases are shown in Tab. 1, with the lowest accuracy for artificial life scientists classifying polarity based on dipole moment for COVID drug discovery.

The algorithmic tradeoff between speedup and accuracy is shown in Fig. 5, showing linear behavior between 80-90% accuracies at an order of magnitude increase in speedup

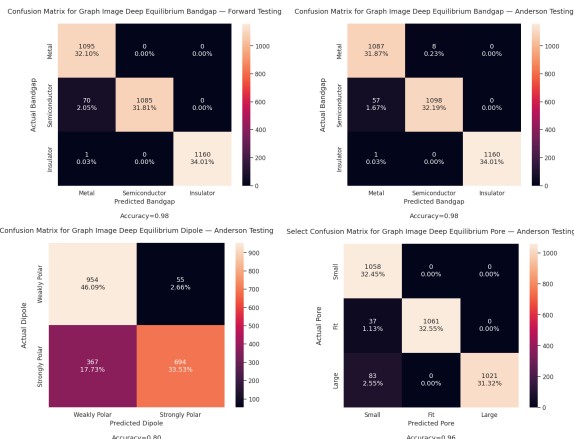

*Figure 4.* Representative confusion matrix deep equilibrium convergence results for testing with (top left) standard forward iteration and (top right) extrapolation for classifying compounds, as well as (lower left) classifying COVID drug dipoles and (lower right) MOF pore sizes.

compared to standard forward iteration when using Anderson acceleration as the forward pass DEQ model solver. A

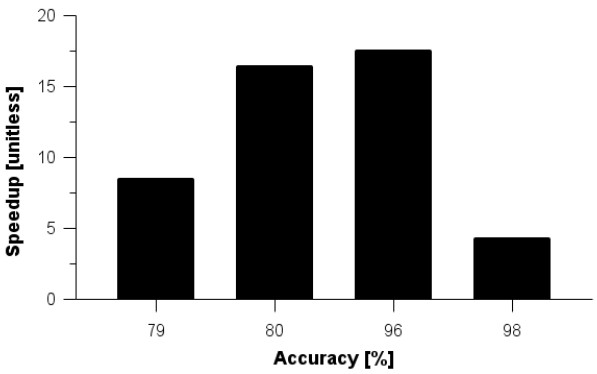

*Figure 5.* Algorithmic trade-off between training speedup and testing accuracy for four use cases with Anderson acceleration plotting the third row of Tab. 2 against the fourth row of Tab. 1.

significant drop in speedup is observed for achieving >95% accuracy up to 98%. These speedups are summarized in Tab. 2.

## 4. Discussion

We demonstrate the effectiveness of learning based on graph images derived from copious density function theory databases (Saal, James E et al., 2013; Kirklin, Scott et al., 2015; Rosen, Andrew S et al., 2022; Isert, Clemens et al., 2022) to predict properties important for life and materials science with application to industrial screening. To achieve accuracies greater than 98%, more data could be included in future studies since these models are trained on about 1,000-10,000 inputs out of these databases.

Single-property classification predictions are shown to be learnable. Future studies should explore multi-objective optimizations for all properties at once, or learning the electron density, energy, forces, and wavefunctions directly from the graph images, from which all properties could be calculated based on standard analysis pipelines.

Algorithmic performance of Anderson acceleration in comparison to standard forward iteration shows different asymptotic accuracies. These could be optimized further by tuning the cosine annealing rate. Furthermore, tuning Anderson acceleration window sizes and iterate-extrapolate ratios could be posed as an AI problem in itself by a grid search for further optimization. This is evident by the representative tunability curve shown in Fig. 2.

The superiority of Anderson acceleration to standard forward iteration in searching for the minimum loss fixed-point, deep equilibrium state is attributed to minimizing a residual over a subspace spanning previous iterations. This appears to overcome the frequent trapping of standard forward itera-

tion in local minima. Anderson accelerations thus delivers some of the benefits expected from second-order methods of machine learning without the overhead of approximating or manipulating Hessian matrices. Its memory-austere operationally uniform character make it eminently suitable for GPUs and for distributed memory implementations.

The effectiveness of graph image node-neighbor representations is attributed to the fact that these are edge-transitive graphs that represent different topologies of drugs, molecules, and compounds, which, in turn, change the properties and bioactivities of these materials in life science applications. Simply by swapping identities of the atoms in these nodes, there are trillions of combinations that could be constructed and trained on for larger datasets for the accuracies required in practical applications. Furthermore, introducing metal alloying, defects, dopants, and other materials engineering methods, could construct even more data to train on. Using graph images enable constructing datasets large enough to build industry-competitive datasets with state-of-the-art accuracies on high-performance computers.

## 5. Conclusion

This work shows that with accelerated deep equilibrium models, artificial life and materials scientists could be constructed for practical industry applications based on first principles theory. A speedup of up to an order of magnitude enables an order of magnitude larger models and/or 90% less computing resources for similar accuracies, paving the way for LLMs, NLSOMs, and foundation models for both training these models and running inferences.

Future work will incorporate larger datasets to build multi-objective optimized models, LLMs, NLSOMs, and foundation models capable of making these types of inferences for drug discovery and biocatalysis at scale, integrating life and materials science in a novel, unprecedented way. With these methods, training models the size of LLMs could be democratized for academic environments without the need to resort to industrial scale computing resources.

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
