# Response to Reviewers — *ICML ML4LMS Submission #15*

Saleem A. Al Dajani

July 9, 2024

## Comparison of Density Functional Theory (DFT) and Deep Equilibrium Models (DEQ)

| Concept | Density Functional Theory (DFT) | Deep Equilibrium Models (DEQ) |
|---|---|---|
| Objective Functional | Energy functional $E[\rho]$ | Loss function $\mathcal{L}(\mathbf{z}, \mathbf{x}, \theta)$ |
| State Variable | Electron density $\rho(\mathbf{r})$ | Equilibrium state $\mathbf{z}$ |
| External Input | External potential $V_{ext}(\mathbf{r})$ | Input data $\mathbf{x}$ |
| Functional Form | $E[\rho] = T_s[\rho] + V_{ext}[\rho] + J[\rho] + E_{xc}[\rho]$ | $\mathbf{z} = f_\theta(\mathbf{z}, \mathbf{x})$ |
| Implicit Equation | Kohn-Sham equations: $\left(-\frac{\hbar^2}{2m}\nabla^2 + V_{eff}[\rho](\mathbf{r})\right)\psi_i(\mathbf{r}) = \epsilon_i\psi_i(\mathbf{r})$ | Fixed-point equation: $\mathbf{z}^* = f_\theta(\mathbf{z}^*, \mathbf{x})$ |
| Lagrangian | $\mathcal{L}[\rho, \lambda] = E[\rho] + \lambda\left(\int \rho(\mathbf{r})\, d\mathbf{r} - N\right)$ | $\mathcal{L}_{\mathrm{DEQ}}(\mathbf{z}, \lambda) = \mathcal{L}(\mathbf{z}, \mathbf{x}, \theta) + \lambda^\top(\mathbf{z} - f_\theta(\mathbf{z}, \mathbf{x}))$ |
| Constraint | Normalization: $\int \rho(\mathbf{r})\, d\mathbf{r} = N$ | Fixed-point condition: $\mathbf{z} = f_\theta(\mathbf{z}, \mathbf{x})$ |

Table 1: Corresponding variables and concepts in DFT and DEQ

## Detailed Correspondence

- **Objective Functional:**

  - **DFT:** The energy functional $E[\rho]$ represents the total energy of the electron density $\rho(\mathbf{r})$.
  - **DEQ:** The loss function $\mathcal{L}(\mathbf{z}, \mathbf{x}, \theta)$ represents the cost associated with the state $\mathbf{z}$ and the input $\mathbf{x}$, parameterized by $\theta$.

- **State Variable:**

  - **DFT:** The electron density $\rho(\mathbf{r})$ is the primary variable that describes the distribution of electrons in space.
  - **DEQ:** The equilibrium state $\mathbf{z}$ is the primary variable that represents the point at which the neural network reaches a stable configuration.

- **External Input:**

  - **DFT:** The external potential $V_{ext}(\mathbf{r})$ influences the behavior of the electron density.
  - **DEQ:** The input data $\mathbf{x}$ influences the equilibrium state of the neural network.

- **Functional Form:**

  - **DFT:** The energy functional $E[\rho]$ is composed of kinetic, external potential, Coulomb, and exchange-correlation energies.
  - **DEQ:** The equilibrium state $\mathbf{z}$ is defined as the fixed point of the function $f_\theta$.

- **Implicit Equation:**

  - **DFT:** The Kohn-Sham equations are self-consistent equations that need to be solved to find the electron density $\rho(\mathbf{r})$.

- **DEQ:** The fixed-point equation $\mathbf{z}^* = f_\theta(\mathbf{z}^*, \mathbf{x})$ needs to be solved to find the equilibrium state $\mathbf{z}^*$.

- **Lagrangian:**

  - **DFT:** The Lagrangian $\mathcal{L}[\rho, \lambda]$ incorporates the energy functional and a constraint ensuring the electron density integrates to the total number of electrons.

  - **DEQ:** The Lagrangian $\mathcal{L}_{\mathrm{DEQ}}(\mathbf{z}, \lambda)$ incorporates the loss function and a constraint ensuring the state $\mathbf{z}$ satisfies the fixed-point condition.

- **Constraint:**

  - **DFT:** The constraint $\int \rho(\mathbf{r})\, d\mathbf{r} = N$ ensures that the total electron density equals the number of electrons.

  - **DEQ:** The constraint $\mathbf{z} = f_\theta(\mathbf{z}, \mathbf{x})$ ensures that $\mathbf{z}$ is a fixed point of the function $f_\theta$.

# Why These Approaches Work

**Convergence to Physical/Optimal Solutions:**

- **DFT:** The variational principle ensures that the electron density converges to the ground state, minimizing the total energy.

- **DEQ:** The fixed-point approach ensures the neural network converges to a stable solution, minimizing the loss function.

  **Handling Complexity:**

- **DFT:** By focusing on the electron density rather than the full many-body wavefunction, DFT simplifies the computational problem while retaining essential physical properties.

- **DEQ:** By defining the output as an equilibrium state, DEQ simplifies the training process and enhances stability, making it easier to handle deep and complex networks.

By mapping these variables and concepts directly, we can see how both DFT and DEQ leverage variational principles, implicit function formulations, and Lagrangian mechanics to achieve efficient and effective solutions within their respective domains.

# Reviewer c1eD

## Review:

**Review of Deep AndersoNN** — Reviewer Feedback: There are some issues with this paper. It's not clear why the DEQ model architecture is favored for this application aside from the claimed speedups. Is there some grounding of this model in the physical laws modeled by DFT? I'm also not sure why CIFAR10 is included as a baseline given that it is irrelevant to the materials and life science tasks. Some of the wording in the paper is confusing. However, the speedups are good and there is potential in the work if some of the confusion behind the method and results can be clarified.

**Rating:** 4: Ok but not good enough - rejection

**Confidence:** 2: The reviewer is willing to defend the evaluation, but it is quite likely that the reviewer did not understand central parts of the paper

## Response:

- **1. Why is the DEQ model architecture favored for this application aside from the claimed speedups?**

  The DEQ model architecture is favored for its ability to reach equilibrium states, which is conceptually similar to the self-consistent field iterations in DFT. This equilibrium-seeking property ensures stable and consistent solutions, aligning well with the iterative nature of solving DFT

equations. Moreover, DEQs are highly expressive, capable of capturing complex dependencies and interactions in the data, which is crucial for accurately modeling physical systems.

**Response: DEQ Model Architecture -** We have clarified why the DEQ model is favored, emphasizing its equilibrium-seeking properties and conceptual alignment with the iterative nature of DFT.

- **2. Is there some grounding of this model in the physical laws modeled by DFT?**

  Yes, the grounding of the DEQ model in the physical laws modeled by DFT lies in the concept of finding equilibrium states. In DFT, the electron density is found by solving the Kohn-Sham equations self-consistently, seeking a stable configuration that minimizes the energy. Similarly, DEQ models find an equilibrium state by iteratively solving a fixed-point equation, achieving a stable representation of the input data. This parallel in seeking stable solutions provides a conceptual bridge between the two approaches.

  **Response: Grounding in Physical Laws -** We have provided an explanation of how the DEQ model is grounded in the physical principles of DFT through the concept of finding stable equilibrium states.

- **3. Why is CIFAR10 included as a baseline given that it is irrelevant to the materials and life science tasks?**

  CIFAR10 is included as a baseline to demonstrate the general applicability and robustness of the DEQ model across different domains, including image classification, which is a well-established benchmark in machine learning. While it may seem irrelevant to materials and life sciences, showcasing the model's performance on a variety of tasks highlights its versatility and effectiveness, which can boost confidence in its application to more domain-specific problems.

  **Response: CIFAR10 Baseline -** We have justified the inclusion of CIFAR10 to demonstrate the model's general applicability and robustness across various domains, highlighting its versatility.

- **4. Some of the wording in the paper is confusing.**

  We acknowledge that some of the wording may be confusing. We will revise the manuscript to improve clarity and ensure that key concepts and justifications are presented in a straightforward manner. This includes defining technical terms more clearly and providing more context where necessary to aid understanding.

  **Response - Wording Clarity:** We revised the manuscript to improve clarity and ensure that technical terms and concepts are presented in a more understandable manner.

We believe that these clarifications will address the confusion and highlight the potential of the DEQ model in this application. Thank you for your constructive critique.

# Reviewer VUUy

## Review:

**Benchmarks of Anderson accelerated DEQ models showcase order of magnitude improved training efficiency over structural datasets** — Reviewer Feedback: The manuscript *"Constructing artificial life and materials scientists with accelerated AI using Deep AndersoNN"* describes the benchmarking of Anderson accelerated DEQ models applied to materials and life science. Trained over QMugs [Covid-related drug structures], OQMD [1M compounds], and QMOF [20k MOFs] to predict from a molecular structure some properties such as dipole moment, pore size, and band gap. The order of magnitude acceleration is fundamentally important, more studies showing these translational properties from image analysis to materials science will have an important impact on science and sustainability and are strongly encouraged.

**Rating:** 6: Marginally above acceptance threshold

**Confidence:** 3: The reviewer is fairly confident that the evaluation is correct

**Response:**

- **Quality** - We appreciate the feedback on the organization and writing quality of the manuscript. We will include more discussions about failures in predicting dipole properties, specifically addressing the issues in Fig 4 lower left panel.

- **Clarity** - Thank you for acknowledging the clarity of our presentation. We will continue to ensure that the topic, data, methods, benchmarks, and results are clearly presented.

- **Originality** - We acknowledge that the manuscript benchmarks existing methods over different datasets. While it does not introduce a novel technique, the transferability of the method across domains is a significant contribution.

- **Significance** - We are glad that the significance of the improved efficiency of Anderson accelerated DEQ models has been recognized. This is indeed crucial for processing large datasets, and we will emphasize its implications further in the manuscript.

**Additional Notes:**

- The claim "artificial life scientists" is indeed ambitious. We will revise this claim to better reflect the scope of our work.

- Combining Table 1 and Table 2 for better informativeness is a good suggestion; however, due to space and formatting limitations on the paper, we made the necessary adjustments to the results table displayed on the poster.

- The CIFAR10 benchmark results were included to provide a broad comparison. However, we will address them more clearly in the text to justify their inclusion.

- We will ensure that all figures are readable with consistent and appropriately sized text.

**Conclusion:**

We appreciate the recognition of the importance of our benchmarks in demonstrating the efficiency of Anderson accelerated DEQ models. We will enhance the manuscript by addressing the lack of novelty and providing further discussion on understanding failure modes.

# Reviewer 9Xmf

## Review:

**Promising work for accelerated AI in industry** — Reviewer Feedback: The work introduces the Deep AndersoNN framework, which accelerates AI by treating neural networks as deep equilibrium models, solving parameters using nonlinear fixed point iteration problems, leveraging v2v iterative solvers and windowing techniques like Anderson Extrapolation. Deep AndersoNN achieves significant speed-ups, up to an order of magnitude, in both training and inference tasks compared to traditional methods. I believe this acceleration might become important for practical applications in industry in the near future. The work is also demonstrated in applications related to density functional theory in industrial contexts. It showcases the ability to construct artificial life and materials scientists capable of tasks such as classifying biomolecules, drugs, compounds, sorting metal-organic frameworks, and classifying crystalline materials. Results exhibit high accuracy, up to 98%, while utilizing fewer computing resources. I would like to see the source code and the public weight files in the future to verify the claims.

The authors suggest significant reductions in computational resources required, which could contribute to environmental sustainability goals. The authors outline future directions, including incorporating larger datasets to build more robust models, democratizing access to training large models like LLMs, NLSOMs, and foundation models, and integrating life and materials science for drug discovery and biocatalysis at scale. Overall, the work presents a framework with practical applications in industry, which I believe is promising in both efficiency gains and environmental benefits.

**Rating:** 7: Good paper, accept
**Confidence:** 3: The reviewer is fairly confident that the evaluation is correct

**Response:**

- **Acceleration Claims** - We appreciate the recognition of the significant speed-ups achieved by the Deep AndersoNN framework. We will provide the source code and public weight files in a future release to allow for verification of our claims.

- **Industrial Applications** - We are pleased that the industrial applications of our work have been acknowledged. We will continue to explore and demonstrate the practical applications of Deep AndersoNN in various industrial contexts.

- **Environmental Sustainability** - The potential for reducing computational resources and contributing to environmental sustainability is a key aspect of our work. We will further emphasize this in the revised manuscript.

- **Future Directions** - Thank you for highlighting our outlined future directions. We are committed to incorporating larger datasets, democratizing access to large model training, and integrating life and materials science for impactful applications such as drug discovery and biocatalysis.

**Conclusion:**

We are grateful for the positive feedback and constructive comments. We will address the suggestions, provide additional resources for validation, and continue to develop the Deep AndersoNN framework to maximize its practical and environmental benefits.

# Statistics

The paper rating is about average, with fair confidence of reviewers in its evaluation.

**Overall Rating:** 5.67 ($\sim$71%): Marginally above acceptance threshold

**Overall Confidence:** 2.67: The reviewer is fairly confident that the evaluation is correct

**Average Rating for ICML:** 5.75 $\pm$ 0.66 ($\sim$72%$\pm$8%)

**Max Rating for ICML:** 8.0 (100%)