# OpenReview forum: "Constructing artificial life and materials scientists with accelerated AI using Deep AndersoNN"
_ICML.cc/2024/Workshop/ML4LMS — ML4LMS Poster_

### Official Review · Reviewer_9Xmf · 2024-05-30
**Promising work for accelerated AI in industry**

**Rating:** 7
**Confidence:** 3

**Review:**

The work introduces the Deep AndersoNN framework, which accelerates AI by treating NNs as deep equilibrium models, solving parameters using nonlinear fixed point iteration problems, leveraging v2v iterative solvers and windowing techniques like Anderson Extrapolation. Deep AndersoNN achieves significant speed-ups, up to an order of magnitude, in both training and inference tasks compared to traditional methods. I believe this acceleration might become important for practical applications in industry in near future. The work is also demonstrated in applications related to density functional theory in industrial contexts. It showcases the ability to construct artificial life and materials scientists capable of tasks such as classifying biomolecules, drugs, compounds, sorting metal-organic frameworks, and classifying crystalline materials. Results exhibit high accuracy, up to 98%, while utilizing fewer computing resources. I would like to see the source code and the public weight files in the future whether it sync with the claim or not.

Authors suggests significant reductions in computational resources required which could contribute to environmental sustainability goals. Authors outlines future directions, including incorporating larger datasets to build more robust models, democratizing access to training large models like LLMs, NLSOMs, and foundation models, and integrating life and materials science for drug discovery and biocatalysis at scale. Overall, the work presents a framework with practical applications in industry, which I believe is promising in both efficiency gains and environmental benefits.

---

### Official Review · Reviewer_VUUy · 2024-06-01
**Benchmarks of Anderson accelerated DEQ models showcase order of magnitude improved training efficiency over structural datasets**

**Rating:** 6
**Confidence:** 3

**Review:**

# Summary:
The manuscript "Constructing artificial life and materials scientists with accelerated AI using Deep AndersoNN" describes the benchmarking of Anderson accelerated DEQ models applied to materials and life science. Trained over QMugs [Covid-related drug structures], OQMD [1M compounds], and QMOF [20k MOFs] to predict from a molecular structure some properties such as dipole moment, pore size, and band gap. The order of magnitude acceleration is fundamentally important, more studies showing these translational properties from image analysis to materials science will have an important impact on science and sustainability and are strongly encouraged.

## Bullet Points:
- **Quality** - The manuscript is well organized and well written. More discussions could be in order, for example, such as in the case of failing in predicting Dipole (Fig 4 lower left panel) properties.
- **Clarity** - As mentioned in the above summary the author clearly presents the topic, data, methods, benchmarks, and results.
- **Originality** - The manuscript includes benchmarking of existing methods over different datasets and while it is important to show that a method can be transferred from one domain to another, it does not introduce a novel technique.
- **Significance** - Benchmarking of the Anderson accelerated DEQ vs. standard DEQ shows an order of magnitude improved efficiency across different datasets and tasks. This is crucial to many tasks that require processing large amounts of data, e.g., biological data as the author clearly discussed.

## Additional Notes:
- The claim “artificial life scientists” is an overstatement. Both data and labels benchmarked in this work are structural and I fail to see how this directly relates to biology. I think the claims should be reconsidered.
- It would be more informative to combine Table 1 and Table 2.
- CIFAR10 benchmark results are not addressed in the text and it is unclear the reason for its introduction in the tables.
- Figures should be made readable. The size of the text in each image is different and in some cases way too small.

## Conclusion:
The Benchmarks presented in this work are important in demonstrating the efficiency of Anderson accelerated DEQ models in learning structural data. At the same time it does not introduce novelty in ML architecture or specific applications and lacks some important discussion for understanding failing modes.

---

### Official Review · Reviewer_c1eD · 2024-06-11
**Review of Deep AndersoNN**

**Rating:** 4
**Confidence:** 2

**Review:**

There are some issues with this paper. It's not clear why the DEQ model architecture is favored for this application aside from the claimed speedups. Is there some grounding of this model in the physical laws modeled by DFT? I'm also not sure why CIFAR10 is included as a baseline given that it is irrelevant to the materials and life science tasks. Some of the wording in the paper is confusing.

However, the speedups are good and there is potential in the work if some of the confusion behind the method and results can be clarified.